# Network Pruning Optimization by Simulated Annealing Algorithm

## Abstract

One critical problem of large neural networks is over-parameterization with a large number of weight parameters. This becomes an obstacle to implement networks in edge devices as well as limiting the development of industrial applications by engineers for machine learning problems. Plenty of papers have shown that the redundant branches can be erased strategically in a fully connected network. In this work, we reduce network complexity by pruning and structure optimization. We propose to do network optimization by Simulated Annealing, a heuristic based non-convex optimization method which can potentially solve this NP-hard problem and find the global minimum for a given percentage of branch pruning given sufficient amount of time. Our results have shown that Simulated Annealing can significantly reduce the complexity of a fully connected neural network with only limited loss of performance.

## 1 Introduction

The successful development of the artificial intelligence methods fueled by deep-learning has made important impact in multiple industries. Elaborate AI algorithm can empower electronic devices to make smart decisions or foresighted predictions. However, the outstanding outcomes come with their own cost, which is the computational complexity in the case of deep-learning methods. In order to obtain a model that can accurately predict the results corresponding to the input data, a neural network should go through initialization, back propagation, gradients update, and inference process step by step. When the deep learning structure is more complicated, there are more parameters and operations involving in these 4 steps. Network compression technique is a relatively new area necessitated by the booming development of deep neural network and increasing computational complexity of machine learning methods. Attracted by the extraordinary performance of recently proposed neural network architectures, there are increasingly more demands on edge devices which possess only limited computational resources. Therefore, various network compression techniques have been tried and studied during these years. The recent mainstream strategies include quantization (Wang et al., 2019), decomposition (Zhang et al., 2015), knowledge distillation (Hinton et al., 2014), and the pruning methods, which is considered as the most effective approach to compress a neural network (Liu et al., 2019b) (Lin et al., 2020a). Among those 4 computational steps of deep learners mentioned above, back propagation and gradients update make up the major part that consumes the most of computing resources. One approach by pruning those gradients that are smaller than a threshold (Ye et al., 2020), significantly reduces the usage of hardware resources and accelerates the whole training procedure. However, the trained model would still have the same parameter size after undergoing the pruned training procedure since the actual weight parameters are not eliminated from the network from computational perspective. Thus, this approach is only helpful for training a network on a powerful computer. In order to actually reduce the total size of a model while inferencing in a mobile electronic device, various research found in the survey (Deng et al., 2020) focus on pruning the weight parameters directly instead of the gradients. These pruning techniques can be element-wise, vector-wise, or block-wise, which can correspond to unstructured or structured pruning. Behind each type of pruning technique, there are multiple criteria being setup differently. One innovative criteria is the combination of generative adversarial network (GAN) and parameters selection (Madaan et al., 2020). By evaluating the similarity of the feature maps before and after pruning process, the importance of certain parameters can be defined.

Except for evaluating the feature map with a designed criteria setup in GAN, the other group of pruning strategies follow heuristic optimization algorithms, where the network is considered as an architecture fine-tuning process. Since a neural network is one type of network composed of both nodes and edges, it would be intuitive to set up the optimization problem in terms of the edge weights and the structure of the network and the pruning strategy. ABC optimization technique is an example that demonstrates the application of a heuristic algorithm on channel-wise network pruning (Lin et al., 2020b). Based on the hierarchical characteristic of the neural network structure, pruning techniques can further be split into both structured and unstructured pruning categories. When a neural network is simulated on computer, the major part of computations would be carried out in matrix operations including addition, multiplication, and dot product. Therefore, in the case of unstructured pruning, the edges or nodes are not truly being eliminated. Instead, an extra mask matrix is designed to simulate the pruning process. Even though the mask matrix can bring up high flexibility in adjusting the network configuration, the model size cannot be reduced significantly since the pruned parameters should still be conserved in the matrix as zero entries. On the contrary, the structured pruning strategies sacrifice some optimality to ensure dimension reduction and avoid sparse but high dimensional matrices. The most representative cases for both unstructured and structured pruning are edge-level and node-level pruning strategies. if one edge of a network is considered as the minimum unit to reduce configuration, it would not affect the existance of all other elements in the network. However, if one node is the minimum unit to be reduce a network, all other edges connected to the selected node will also need to be erased.

In choosing pruning strategies, the main factor is to determine which part of the minimum unit (edges, nodes, ... etc) is more important than the others. The existing pruning approaches do not consider the optimality of the resulting network structure in any means. The optimality under question has two aspects to it: the branch weights and the branch connections. In this work, we aim to optimize the branch connections with the constraint of pruning percentage, That is, we question the optimality of the structure obtained by pruning. In this paper, we augment the network pruning operation with a structure optimization algorithm. For this task we propose simulated annealing (SA) algorithm. By considering the permutation of the pruned configurations as a finite Markov chain, the best remaining configuration can be cleverly searched by Simulated Annealing. Different from the other pruning approaches which defined additional fancy objective function and constraints, our approach directly takes the loss function of a neural network as an objective function. The optimized network is guaranteed to be the best permutated configuration under the fixed parameters. In addition, since the loss function is used to both configuration optimization and parameters update with back propagation, the optimal configuration and the remaining parameters can further be trained to fine-tune the parameters seamlessly.

## 2    RELATED WORKS

With the increasing applications of deep learning, there are more and more demands for the portability of the DNNs and two lines of research evolved over the years to reduce the computational and storage requirements of DNNs. The first line of research focused on reducing the storage requirements of DNNs; this is achieved by formulating the problem as source coding or data compression problem and making use of existing quantization methods. Examples of this approach include: low bit quantization (Hubara et al., 2018), multi-level quantization (Xu et al., 2018) and clustering based quantization (Choi et al., 2017). More elaborate source coding schemes are also employed such as Huffman encoding in (Han et al., 2016) and trellis coded quantization (Haase et al., 2020) and entropy based coding (Wiedemann et al., 2020).

The other line of approach aims at reducing the number of parameters in the neural network by making it more sparse which is achieved by pruning. The pruning approach was motivated by the very early (pre-deeplearning) works of LeCun et al (LeCun et al., 1989) and Hassibi and Stork (Hassibi & Stork, 1992), which showed that neural networks can still work well when many edges or nodes are pruned.

### 2.1    NETWORK PRUNING TECHNIQUES

As exemplified in *deep compression* (Han et al., 2016), one simple yet effective pruning criteria is to select those weights with absolute magnitude smaller than a threshold. The process is executed

based on a pretrained network. Once the small weights are pruned, the remaining weights will be fine-tuned by training with the dataset again. Various pruning algorithms differ in their choice of where they apply the threshold such as the output of activation function or the gradient of each weight parameter or the magnitude of the weights.

It has also been reported in the deep compression approach (Han et al., 2016) that pruning the network gradually by picking up only a small fraction of total weights parameters at one time can help to maintain the performance of the pruned network. However, the sparse network still occupies a lot of space in a computer because the pruned weights are all represented by zeros. In order to truely eliminate the parameters from the network, (Chen et al., 2020) proposed tight compression which converts the resulting large sparse matrix after pruning to a smaller but dense matrix by moving non-zero members to matrix locations with zeros. These moves are not done in a deterministic manner but instead via stochastic heuristic algorithm namely simulated annealing.

## 2.2 NEURAL ARCHITECTURE SEARCH (NAS)

Despite the conceptual simplicity and ease of application of pruning, it is a valid question to ask whether the resulting sparse networks are optimal. That is, do the resulting networks structures (with the weights assigned after tuning) provide globally optimal results for given data sets. The straight answer to this question is that given the pruned structures, the tuned weights provide local optimals but this observation tells us nothing about the optimality of the structures. This issue has rarely been discussed in the pruning literature. There are some work, however, which consider neural architecture search mostly out of the context of pruning.

The researchers have considered the structure of a DNN as composed of operational blocks. These blocks include convolution, activation functions, normalization functions, regularization functions, and so many. According to the various combinations of these blocks, different structures of DNN can be constructed. The constructing process can be designed to progress while the model is being trained. By selectively searching all possible queued operations, the optimal configuration can also be determined. When there are more types of operations, or the operation queue is getting longer, the total number of possible states will grow exponentially. This could be a severe problem in neural architecture search especially when it follows grid-search strategy (Zoph & Le, 2017). More effective ways to search for the optimal combination of blocks and permutation of active branches were also proposed by Zoph (Zoph et al., 2018), which stacks the basic blocks sequentially and gradually instead of directly finding the optimal whole network architecture. One probabilistic model-based dynamic optimization method tried to implement a penalty term to the objective function so that the gradients can lead to a better convergence based on a generic framework, which determines a network structure by a vector (Saito & Shirakawa, 2019) (Shirakawa et al., 2018). The other perspective of structure search is to construct a computational graph by including all potential operational blocks (Liu et al., 2019a) (Pham et al., 2018). These approaches all focused on the macro scale of structure optimization without considering the importance of each parameters contained in an operation. Even though the optimal combination of blocks are determined, these structures still may have redundant parameters for weights pruning.

Optimization of structure in micro scale avoiding redundant branches seems to be rare in the literature. One exception is (Nowakowski et al., 2018) which utilizes genetic algorithms for the structure optimization task. Our proposed methodology, is similar in spirit to this work, and is detailed in the next section.

## 3 NETWORK PRUNING WITH SIMULATED ANNEALINGS

In this work, we propose a heuristic non-convex optimization algorithm, namely Simulated annealing (SA) for structure optimization of partially connected neural networks after pruning. The choice of Simulated Annealing has been motivated by the success of the algorithm in various problems involving network/graph structures with large number of configurations and complicated cost surfaces with various local minima (Kuruoglu & Arndt, 2017), (Kuruoglu & Ayanoglu, 1993), (Liu et al., 2021).

SA is motivated by the annealing process in solid state physics, which aims to place the electrons in a solid at their lowest energy states achieving lowest possible energy configuration (Metropolis

et al., 1953). Similar to typical discrete optimization problems, SA is given a finite set containing all possible configurations $\mathcal{C}$ from a fully connected neural network $N(\cdot)$ and a loss function $\mathcal{L}(\cdot)$, and looks for $\mathbf{c}^* \in \mathcal{C}$ such that $\mathcal{L}(\mathbf{c}^*)$ is minimized. The loss function defined in the optimization problem of neural network is the energy function defined in solid state physics. Different from the panaroma of gradient descent algorithms, which are very likely to be trapped in a local optima, Simulated Annealing algorithm determines the global optima by doing a search in the solution space following a Markov chain random walk.

Simulated Annealing is a general algorithm which can be applied to various problems, which is documented by its success in vastly different application areas; however as in any general algorithm, it requires the design of its components by the user according to the application.

The success of SA depends on careful design of three crucial mechanisms: state neighbourhood structure selection, acceptance-rejection criteria of proposals and the cooling schedule.

## 3.1 CHOICE OF STATE NEIGHBORHOOD STRUCTURE

The choice of neighborhood structure dictates the possible moves the random walk can make; hence it affects significantly convergence rate. A too conservative neighbourhood structure would make the exploration of the solution space very slow while a too liberal choice may make the random walk jump over important minima and will not learn enough turning it into a blind random search algorithm on all possible configurations. However, a conservative or liberal neighbourhood structure does not always provide the same level of disadvantage or advantage. At the beginning of annealing process, big changes between neighborhoods can be helpful for a quick (and coarse) search of the solution space. When a potentially interesting part of the solution space has been reached, smaller moves in a conservatively defined neighbourhood would be more beneficial there are more space being looked around, small changes turn out to be more beneficial to capture the global optimum. On the other hand starting with too conservative neighborhoods would have made the algorithm stuck in local minima nearby the starting configuration.

For the network structure optimization problem, we define the states in the solution space as the configuration of branches connecting nodes in one layer to the next one. Equivalently, we consider a mask matrix $M_l$ which is filled with 1's at locations $(i, j)$ where there is a surviving branch between the nodes $i$ and $j$ after pruning and with 0's at locations $(i, j)$ where there are no surviving branches.

The neighbouring states are defined to be new configurations obtained by moving one (or more) branch(es) originally at location $(i, j)$ to $(i, k)$. Equivalently, can be visualised as the move of one $1$ from matrix location $(i, j)$ to another position on the same row $(i, k)$. The choice of a single branch replacement corresponds to a conservative neighbourhood structure. It can be made more liberal by considering more than one replacements.

## 3.2 ACCEPTANCE-REJECTION RATIO

Simulated Annealing algorithm can jump out of local optimum is since it occasionally accepts a new state that increases the outcome of the loss function $\mathcal{L}(\cdot)$ . This mechanism reflects an analogy with the electron dynamics in solid state physics, when the electron has enough thermal energy it can jump over barriers and end up in higher energy states and hence can avoid some local energy minima. As in the case of electron dynamics, SA adopts a Boltzmann distribution to decide accepting or rejecting a move or a step of the random walk:

$$\mathcal{P}_b = \min\left(1, \exp\left(\frac{-\Delta\mathcal{L}(\cdot)}{k \cdot T}\right)\right) \tag{1}$$

At every move, $-\Delta\mathcal{L}(\cdot) = \mathcal{L}(M) - \mathcal{L}(-M')$ is calculated, if it is larger than $0$, the new state is accepted, otherwise a uniform number is generated between $[0, 1)$ and compared with $\mathcal{P}_b$. If $\mathcal{P}_b$ greater it is accepted, otherwise it is rejected.

## 3.3 CONVERGENCE

An important question is whether this accept/reject scheme and the choice of neighbourhood structure ensure convergence. The answer to this question is well known in the simulated annealing and Markov chain theory literature (van Laarhoven & Aarts, 1987).

A (finite-state) Markov chain converges to a unique stationary distribution only if it has two fundamental properties: aperiodicity and irreducibility or equivalently ergodicity. These properties are not easy to check generally and therefore a stronger property is used, namely detailed balance.

For our problem, the detailed balance condition can be expressed as:

$$P_{\mathbf{c}}(\boldsymbol{M}) \cdot P(\boldsymbol{M'}|\boldsymbol{M}) = P_{\mathbf{c'}}(\boldsymbol{M'}) \cdot P(\boldsymbol{M}|\boldsymbol{M'}) \tag{2}$$

$P_{\mathbf{c}}$ indicates the probability that the configuration is sampled under the representation of mask matrix $\boldsymbol{M}$, and $\boldsymbol{M'}$ indicates the first order neighbor configuration of the Markov chain after $\boldsymbol{M}$.

It can be shown that the accept/reject mechanism using the Boltzmann function in Equation (1) ensures that the detailed balance condition and therefore converges to the unique stationary distribution (van Laarhoven & Aarts, 1987). This property is shared with the Markov chain Monte Carlo (MCMC) algorithm which aims to obtain the posterior distribution of model parameters. Both algorithms construct Metropolis loops via random walks over a Markov chains satisfying the detailed balance condition. The difference is that Simulated Annealing algorithm employs several Metropolis loops with decreasing temperature parameter $T$ in Equation 1. Hence, posterior distribution approximated by every Metropolis loop gets more and more peaked at the maximum of the stationary distribution. In this way, the parameter values that maximize the stationary distribution of the Markov chains is obtained.

### 3.4 Cooling Scheme and Hyperparameters

Another design issue affecting the convergence rate of SA is the cooling scheme. It is known that the convergence is guaranteed in the case of logaritmic cooling, however this requires infinitely slow cooling. Instead, most users prefer geometric cooling

$$T_{n+1} = \eta T_n.$$

The important parameters to be set are the initial temperature ($T_{init}$), the cooling rate ($\eta$) between Metrolopis loops, and the Metropolis loop length ($\mathcal{MLL}$). The temperature decrease controls the acceptance-rejection probability for a new state that has higher loss value. If the temperature is high, the worse states are more likely to be accepted. As the temperature is reduced increasingly, the worse states are more likely to be rejected. The decreasing rate is the main factor responsible for handling the speed of the whole annealing process. However, a faster annealing process does not guarantee the convergence of a optimization problem. The objective function can be minimized only when the decreasing speed is slow enough. In our work, we setup the initial $T$ as 10 and $\eta$ as 0.98. The third parameter $\mathcal{MLL}$ should be large enough for the Markov chain to converge to the stationary distribution and small enough not lose time unnecessarily beyond convergence.

### 3.5 Heuristic Attention Mechanism

Attention mechanism aims at finding the importance of elements contained in a network. If the element is more important than the others, its weighted factor, which is represented as a number, will be larger. Generally speaking, the element can be anything except for weight parameters. For example, the elements defined in SqueezeNet (Hu et al., 2018) are the channels in 2D convolutional operation. By including the weighted factors to multiply by each channel of the feature maps in the forward pass, back propagation can be applied to learn the importance of each element. Different from the typical learning method to achieve attention mechanism, Simulated Annealing finds the important elements by trial-and-error with Boltzmann probability. However, unlike the continuous values given from the derivatives using back propagation, the weighted factors computed by our approach would be discrete numbers $\{0, 1\}$ where 0 and 1 indicate that the elements are ignorable and crucial respectively. These two types of values are actually the fundamental elements composing the mask matrix ($M$). Being a new incorporation between SA and network pruning, we call our method as the heuristic attention mechanism. The details are described in **Algorithm 1** where a network takes a dataset $\mathcal{T}$ and a set of mask matrices as inputs. The $c$ marked in the bracket indicates the $c$-th type of configuration represented by $M$.

### 3.6 PERMUTATION AFTER EDGES PRUNING

According to **Algorithm 1**, the mechanism starts by deciding how many less important elements to eliminate randomly. Even though the elements might be eliminated in one loop, they still have chance to be recovered back during the random walk with the acceptance-rejection mechanism. Starting from a sparser network, an optimal sub-configuration can be determined iteratively by Simulated Annealing algorithm. If the total number of elements is larger, the corresponding Markov chain must be longer so that a wider space is explored. The longer length indicates that the initial temperature should be higher and the decreasing rate should be closer to 1 so that SA has more time to search for the whole space.

While the network performance will strictly go down as there are increasingly more parameters being erased, the optimal permutation can always be determined under the pruning percentage setup in advance. Different from (Rere et al., 2015) in which applied SA to update weights parameter directly, we implemented it to decide which edges should be kept connected and which should not be in each iteration. In order to change connections in a more convenient way, the actual objects that we are fine-tuning would be the mask matrices. The whole optimization process starts from a network trained using back propagation approach.

$$\min_{w_i^*, b_i^*} \mathcal{L}\Big(N(\mathcal{T}_{train}, |\{w_i, b_i\}_{i=1}^k, \mathcal{T}_{train}^{labels}\Big) \tag{3}$$

After getting the optimized weight parameters, we erase a small fraction of total weight parameters and start the permutation process. These two steps can both be fulfilled by manipulating the mask matrices. According to the pruning objective, which is minimizing the same loss function $\mathcal{L}(\cdot)$ applied in Eq.(3) by fine-tuning network configuration, the objective function can be formulated as follows.

$$\underset{M_i^{(c*)}}{argmin} \mathcal{L}\Big(N(\mathcal{T}_{train}^{data}, \{M_i^{(c)}\}_{i=1}^k)|\{w_i^*, b_i^*\}_{i=1}^k, \mathcal{T}_{train}^{labels}\Big) \tag{4}$$

The objective function Eq.(3) optimizes the configuration under a fixed preset fraction of randomly pruned network. When the new configuration is finalized, the remaining weight parameters should be fine-tuned again to make sure the network is stabilized. During the pruning process, the parameters are not actually being erased. Instead, the numbers are masked with zeros. The difference between mask matrices and dropout operation is that the gradient conducted by back propagation cannot update the parameters being masked by zeros, but can still update the parameters in dropout case.

$$\underset{w_i^*, b_i^*}{Min} \mathcal{L}\Big(N(\mathcal{T}_{train}, \{M_i^{(c)}\}_{i=1}^k)|\{w_i, b_i\}_{i=1}^k, \mathcal{T}_{train}^{labels}\Big) \tag{5}$$

Therefore, the optimal $w_i^*$ and $b_i^*$ are specifically indicating those remaining parameters. In order to evaluate the performance and the effectiveness of **Algorithm 1** in the field of network pruning, we implement it to a simple and shallow fully connected network illustrated in **Figure 1** which contains only 2 hidden layers.

In both cases of one-shot and gradual pruning strategies, it basically follows the same framework composed of Eq.(3, 4, 5). Once there are more percentage of elements being eliminated, both Eq.(4) and Eq.(5) will need to be executed again in order to keep getting the optimal configuration (permutation). The efficacy of Eq.(4) can be obviously found in the following one-shot pruning experiment and the importance of Eq.(5) can be observable by comparing the results between these 2 strategies.

Here, we would like to note a shortcoming of this algorithm whether be one-shot or gradual pruning. The algorithm optimizes the structure for a given set of weights which were trained for the unpruned network. The optimal weights for a new configuration is not the same with weights taken from the unpruned network. Ideally, the Algorithm should train the new structure weights at each new structure proposal, however this is avoided due to computational complexity increased by the length of the Metropolis loop. Although the network is trained finally when the metropolis loop has ended, of course the decisions that were made by Accept/Reject ratio are affected by the untrained weights. Our aim is to show that even for this suboptimal case improvements can be obtained motivating future research into structure optimization.

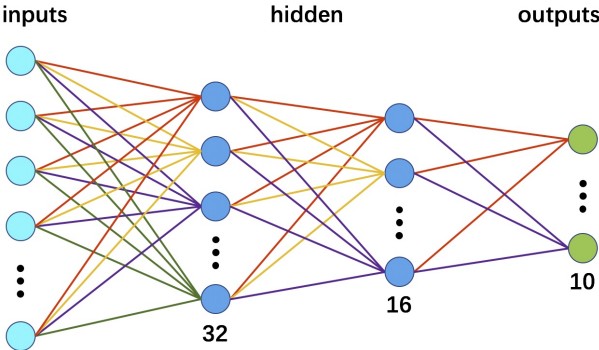

Figure 1: Configuration of a shallow network. This network is designed to evaluate the effectiveness of SA in the field of network pruning. In order to keep things simple, only the edges in hidden layers will be pruned.

### 3.7 ONE-SHOT PRUNING

One-shot pruning is a direct and time-efficient approach to recompose a network. Once the optimal parameters $w^*$ and $b^*$ are obtained in Eq.(3) with back propagation, the pruning and permutation process can be executed. These operations can be considered as some interference to a stable network. The more changes to the configuration, the higher loss value would be computed from the new configuration. This trend can be clearly observed in different percentage of one-shot pruning. Since the pruning process is only executed once, the above equations, Eq.(3)(4) and (5), will only be executed once, where the details are listed in **Algorithm 2**. In order to make sure that the permutation process done in **Algorithm 1** can work properly, the pruning process is necessary because both connected and disconnected edges will exist in a network.

However, the efficiency of one-shot pruning comes at a cost, which could have been avoided by the gradual adaptation process. Its performance would significantly drop especially when there are increasingly more parameters being pruned. This phenomenon can be observed from the cases by taking both *MNIST* and *FASHION* datasets as $\mathcal{T}_{train}$ for our network in figure 2.

### 3.8 GRADUAL PRUNING

Gradual pruning is widely known to be a better solution to trim a network rather than pruning at once. Instead of executing the above equations only once, gradual pruning executes them multiple times to gradually decrease the total number of network connections. By doing this, the change in new configuration will get more time to fine-tune the remaining weight parameters so that the new loss value will keep close to the original loss value. In the case of gradual pruning, a small fraction should be setup in advance. In this paper, we applied 2%, 5%, and 10% in total.

In the beginning of the **Algorithm 2**, the fully connected network should also be well trained. For the next steps being done iteratively, a small preset fraction of weight parameters would be masked out randomly so that the network will contain both connected and disconnected types of edges. After getting an optimal permutation, the network will further be trained for few more epochs to certify weight parameters still remain at stable state. By repeatedly mask out more and more weight parameters, a final sparse network can be obtained.

In order to observe the effectiveness of **Algorithm 2**, we also designed two baseline strategies. The first one is to randomly select weight parameters and the second one is to select the minimum $\mathcal{K}$ amount of absolute weight parameters $(min\mathcal{K})$ inspired by deep compression (Han et al., 2016). When there are gradually less remaining parameters, the statistics of total weight parameters are changed slightly because of the fine-tuning process in Eq.(5). This step plays an important role to prevent the performance of the network from dropping significantly when high percentage of weight parameters are pruned. In addition, optimization with back propagation has strong adaptability regardless of the pruning strategy. In the case of $min\mathcal{K}$, more weights were updated into larger

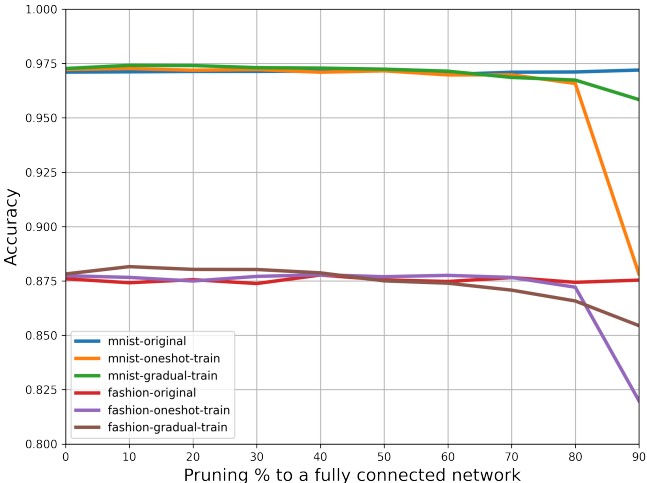

Figure 2: The comparison between three cases from 2 different datasets: original network, one-shot pruned and trained network, and gradual pruned and trained network. Each result is obtained by computing the mean of 5 independent and identical experiments.

absolute values and the distribution were transformed into a slight sharpened shape. As for the case of SA, the final distribution basically remained at similar shape and the weight parameters at different scales were pruned relatively evenly.

## 4 EXPERIMENTS

According to the procedures of various pruning strategies introduced above, the remaining weight parameters and the corresponding configuration can be really different. In order to evaluate the performance of the pruned network at different sparsity levels, two datasets were applied to the experiments individually in this paper. The neural network applied in our experiment is a simple structure, containing only 2 hidden layers with 32 and 16 nodes respectively. The illustration has been shown in Figure 1. Simulated annealing algorithm has shown its capability to find out the optimal configuration under the situation that all parameters are fixed. To solve a more complex optimization problems, more random walk is guarenteed to increase the performance of the final result. The experiments under 0, 1, 5, 10, 20, 50, and 100 $\mathcal{MLL}$ have all been included in this paper. However, better optimization results come at the cost of time complexity of **Algorithm 1**. According to our experiments, 50 $\mathcal{MLL}$ has been seen to be good compromise to get both good performance and tolerable time consumption.

### 4.1 ONE-SHOT PRUNING

The disadvantage of one-shot pruning strategy is that the accuracy will significantly drop when there are too many edges being pruned at one time. The trend has been illustrated in *figure 2*. Before achieving the threshold of 80% sparsity level, the remaining network can still be recovered by applying back propagation to Eq.(5). Both *figure 5* and *figure 6* have shown the changes of network performance from the time when the network is randomly pruned to the time when the configuration is permuted and fine-tuned again. As there are more edges being pruned, the performance gap between the fine-tuned network and the original fully connected network is getting larger.

With the help of SA, our experiments have shown that the weight parameters fine-tuning process in Eq.(5) can be omitted when the pruning percentage is not high. It would become much more convenient without training the network in a GPU and the pruning process could even be executed on edge devices. In the case of shallow network used in this paper, SA with 100 $\mathcal{MLL}$ can almost

recover the performance of the network being pruned 80%. In addition, when the Metropolis loop length is big enough, the optimized configuration is guarenteed to outperform the other configuration obtained from $min\mathcal{K}$ strategy, which is suggested in deep compression (Han et al., 2016). The same trend can be observed in both datasets.

## 4.2 GRADUAL PRUNING UNDER DIFFERENT MLL

In the case of gradual pruning, there is one more hyperparameter included in the experiment, which is the fixed percentage being used to erase edges iteratively. Accordingly, we get one more dimension to compare the experimental results. Similar to the success obtained in one-shot pruning strategy, Simulated Annealing algorithm also defeats the other two approaches in gradual pruning when the Metropolis loop length is large enough. The accuracy of the network at different time steps has been shown in *figure 7* and *figure 8*. The network strictly undergoes the optimization process of the three equations Eq.(3, 4, 5). The trends of network performance were the results after being permutated and trained to fine-tune weight parameters again. The trends indicate that the network being pruned more percentage at one time during each iteration will decrease the performance faster. In order to maintain the performance of a sparser network, gradual pruning has been experimentally proved to be an valid strategy.

In addition, the fine-tuning step is capable of pushing the performance of pruned network to a higher level, where the accuracy would be increased up to 1%. This trend is observable before the hidden layer is masked less than half. According to the *figure 7* and *figure 8* at different masked percentage for the hidden layer, SA with large metropolis loop length (50+ $\mathcal{MLL}$) always outperforms the others. The small amount of accuracy increment and large $\mathcal{MLL}$ are proved that they can slow down the decrease in network accuracy especially when majority of network edges are pruned. Unlike the attention mechanism applied in SqueezeNet (Hu et al., 2018) quantifying the importance factor directly to the training process optimization with back propagation, SA needs huge amount of time to locate the global optima by the Boltzmann accept reject mechanism. This is the main disadvantage of SA.

## 5 CONCLUSION

The most important observation of our work is that direct pruning does not necessarily lead to optimal network structures and changing the network structure can give us better results even in a weight setting which was not optimized for the new configurations. It has also been experimentally shown that the weight parameters with small magnitude are not certain to be the less important parameters. With the help of SA and back propagation to update weight parameters, a more lightweight configuration can be obtained without sacrificing its performance. The main shortcoming of the presented algorithm is that for each new structure proposal during SA metropolis loop, weight training using backpropagation was not performed. Obviously, the resulting search is suboptimal. Unfortunately doing weight training hand in hand with structure training turned out to be computational prohibitive and impossible for portable devices. However, the results obtained even with this suboptimal scheme show the potential of structure optimization and motivate for further research into network optimization. Speeding up SA so that full optimization can be carried is the objective of our current research.

The other advantage of our work is that the pruning and permuting processes with Simulated Annealing algorithm involves only forward pass. It indicates that RAM space contained in a computer can be hugely saved, and the pruning process can therefore be executed on a less powerful computer. In addition, our work has strong extensibility to other types of network pruning. In this paper, we took edges as the basic element to fine-tune the configuration, which generated a very long finite Markov chain. However, being an intelligent state space exploration method, the Simulated Annealing algorithm can be coded also for other types of basic elements, such as node and Convolution filter. By asking SA which filter to erase instead of which edge, the time complexity can be reduced to acceptable levels in the pruning process of complex network with the help of GPU.

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

# A APPENDIX

---

**Algorithm 1** Heuristic Attention Mechanism using SA

---

**Input**: $N(\mathcal{T}_{train}, \{M_i^{(c)}\}_{i=1}^k), M_i^{(c)} = \{0,1\}^{in \times out}, c \in \mathbb{R}$
**Parameter**: $T_{init}, T_{min}, \eta, k, \mathcal{MLL}$
**Notes**: $\mathcal{MLL}$ stands for "Metropolis-loop Length"
**Output**: $N(\mathcal{T}_{train}, \{M_i^{(c^*)}\}_{i=1}^k)$

1: Let $t = T_{init}$.
2: **while** $t > T_{min}$ **do**
3:     **for** i in $1 \sim \mathcal{MLL}$ **do**
4:         Compute $Loss = \mathcal{L}(N, \mathcal{T}_{train}^{labels})$
5:         Randomly pick 1 connected link in hidden layer.
6:         Randomly pick 1 disconnected link in hidden layer.
7:         Change connection status $M_{hidden}^{(c')} \leftarrow M_{hidden}^{(c)}$
8:         Compute $Loss' = \mathcal{L}(N'(M_{hidden}^{(c')}), \mathcal{T}_{train}^{labels})$
9:         **if** $Loss' < Loss$ **then**
10:           Accept new configuration $N \leftarrow N'$
11:         **else**
12:           Random $r(0,1)$
13:           **if** $r > \exp \frac{-(Loss' - Loss)}{k \cdot T}$ **then**
14:             Accept new configuration $N \leftarrow N'$
15:           **else**
16:             Remain the original network configuration
17:           **end if**
18:         **end if**
19:     **end for**
20:     $t \leftarrow \eta \cdot t$
21: **end while**
22: **return** $N(\mathcal{T}_{train}, \{M_i^{(c^*)}\}_{i=1}^k)$

---

**Algorithm 2** One-shot Pruning with SA (OSPSA)

---

**Input**: $N(\mathcal{T}_{train}^{data}, \{M_{i|r_i}^{(c)}\}_{i=1}^k), \boldsymbol{M}_i^{(c)} = \{0,1\}^{in \times out}$
**Parameter**: $\boldsymbol{r} \in (0,1]^{1 \times k}$
**Notes**: $r$ indicates the ratio of remained weights parameter.
**Output**: $N(\mathcal{T}_{train}^{data}, \{M_{i|r_i'}^{(c^*)}\}_{i=1}^k)$

1: Let $p = 50\%$
2: $w_i^*, b_i^* \leftarrow Minimize \, \mathcal{L}(N, \mathcal{T}_{train}^{labels})$ in Eq.(3)
3: $r_i' \leftarrow r_i - p$
4: $N'(\{M_{i|r_i'}^{(c^*)}\}_{i=1}^k) \leftarrow$ **Algorithm 1**$(N)$
5: $w_i^*, b_i^* \leftarrow Minimize \, \mathcal{L}(N', \mathcal{T}_{train}^{labels})$ in Eq.(5)
6: **return** $N'(\mathcal{T}_{train}^{data}, \{M_{i|r_i'}^{(c^*)}\}_{i=1}^k)$

---

---

**Algorithm 3** Gradual Pruning with SA (GPSA)

---

**Input**: $N(\mathcal{T}_{train}^{data}, \{M_{i|r_i}^{(c)}\}_{i=1}^k), M_i^{(c)} = \{0,1\}^{in \times out}$
**Parameter**: $\mathbf{r} \in (0,1]^{1 \times k}$
**Notes**: $r$ indicates the ratio of remained weights parameter.
**Output**: $N(\mathcal{T}_{train}^{data}, \{M_{i|r_i}^{(c^*)}\}_{i=1}^k)$

 1: Let $p = 10\%$
 2: $w_i^*, b_i^* \leftarrow Minimize\ \mathcal{L}(N, \mathcal{T}_{train}^{labels})$ in Eq.(3)
 3: **while** $r_i > p$ **do**
 4: $\quad r_i' \leftarrow r_i - p$
 5: $\quad N'(\{M_{i|r_i'}^{(c^*)}\}_{i=1}^k) \leftarrow \textbf{Algorithm1}(N)$
 6: $\quad w_i^*, b_i^* \leftarrow Minimize\ \mathcal{L}(N, \mathcal{T}_{train}^{labels})$ in Eq.(5)
 7: $\quad r_i \leftarrow r_i'$
 8: $\quad N \leftarrow N'$
 9: **end while**
10: **return** $N(\mathcal{T}_{train}^{data}, \{M_{i|r_i}^{(c^*)}\}_{i=1}^k)$

---

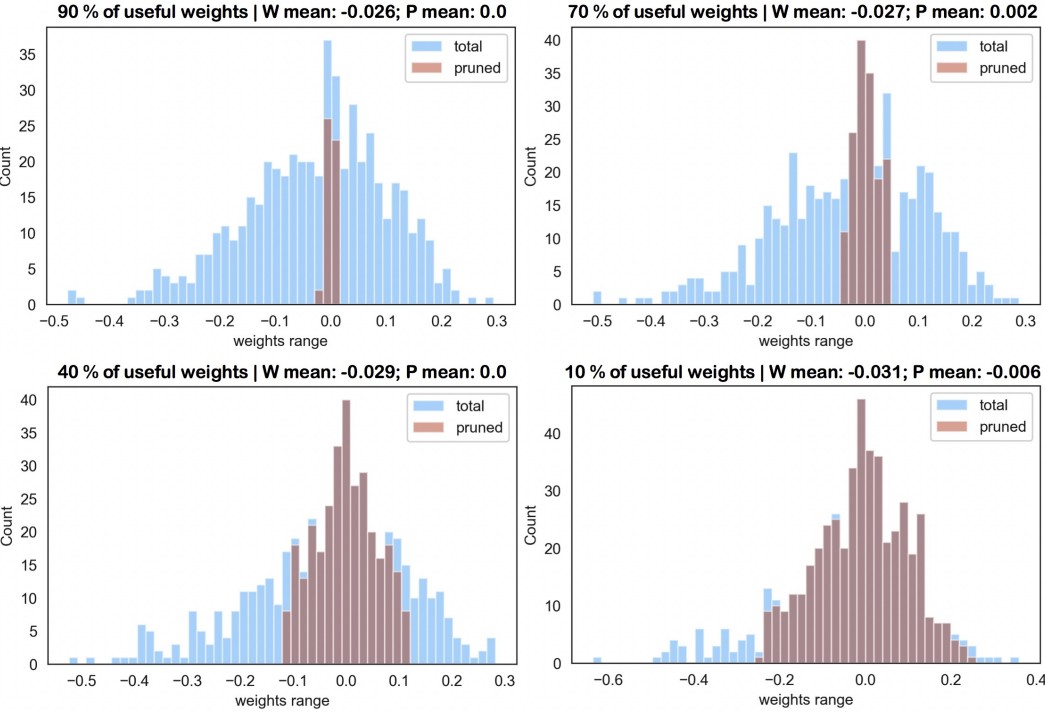

Figure 3: The histogram of total weight parameters under the process of gradual pruning using $min - \mathcal{K}$ strategy as $p = 10\%$. It will always determine those weights which has lowest absolute value.

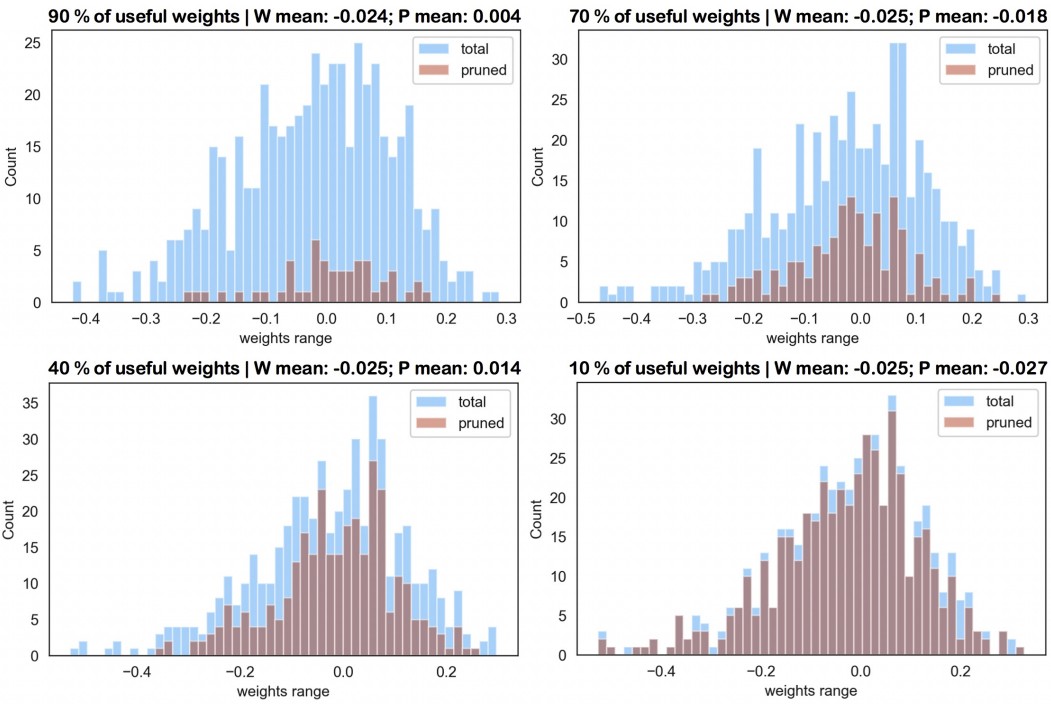

Figure 4: The histogram of total weight parameters under the process of gradual pruning followed by **Algorithm 1** as $p = 10\%$. Those less effective weights will be determined by simulated annealing algorithm iteratively.

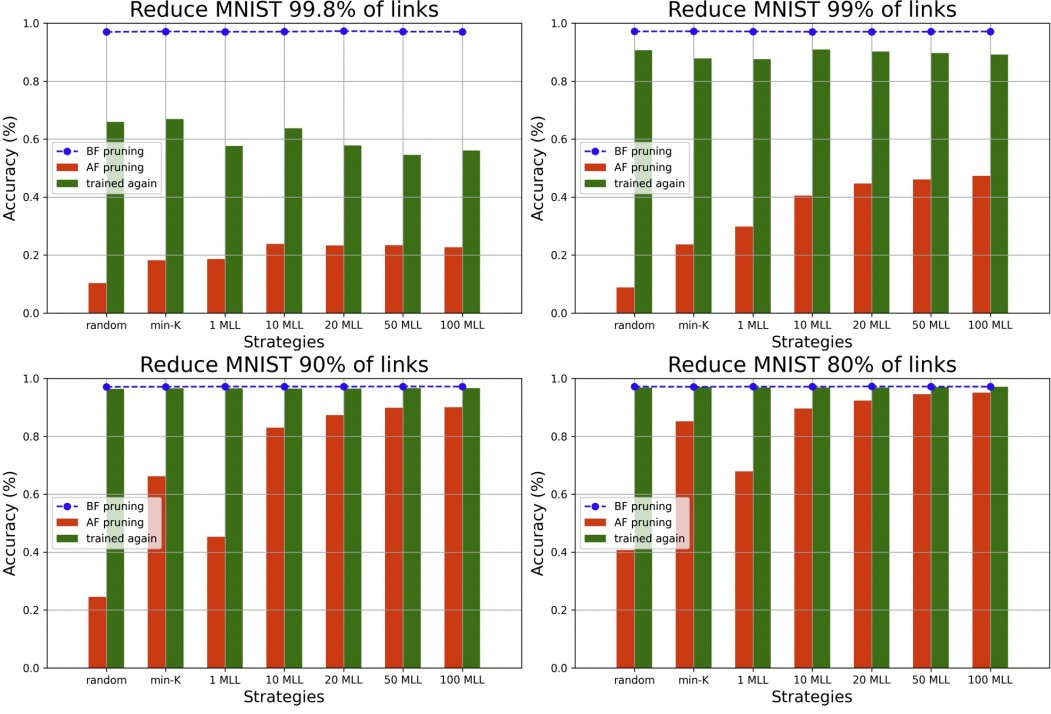

Figure 5: Performance changes under various pruning percentage and different permutation strategies. The network is well trained by MNIST dataset.

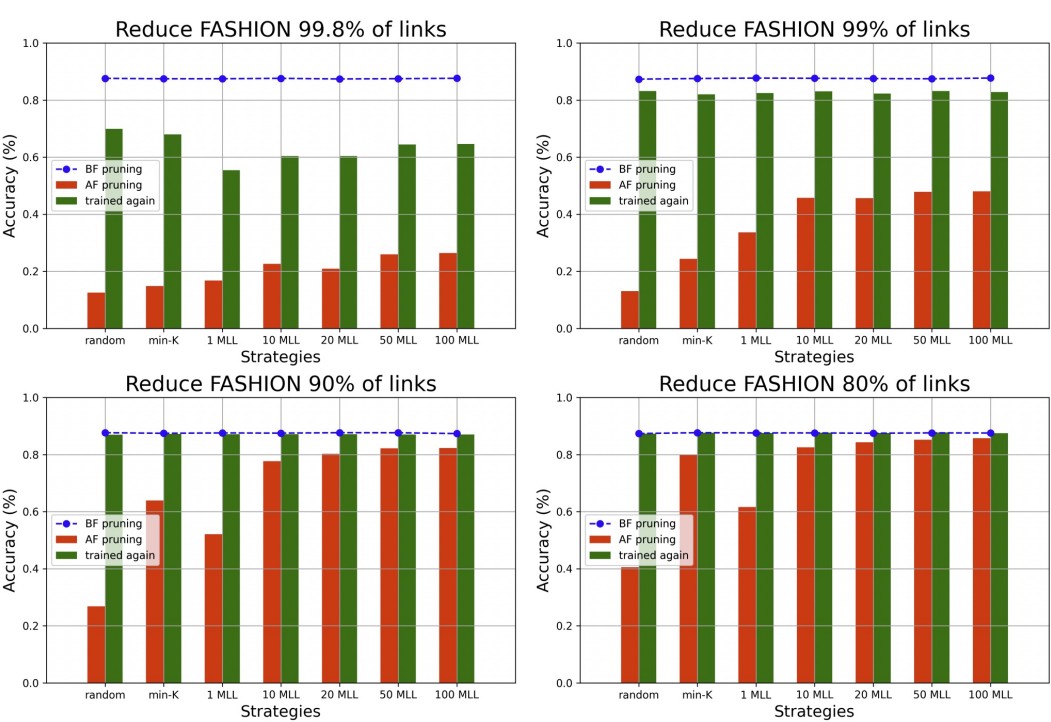

Figure 6: Performance changes under various pruning percentage and different permutation strategies. The network is well trained by FASHION dataset.

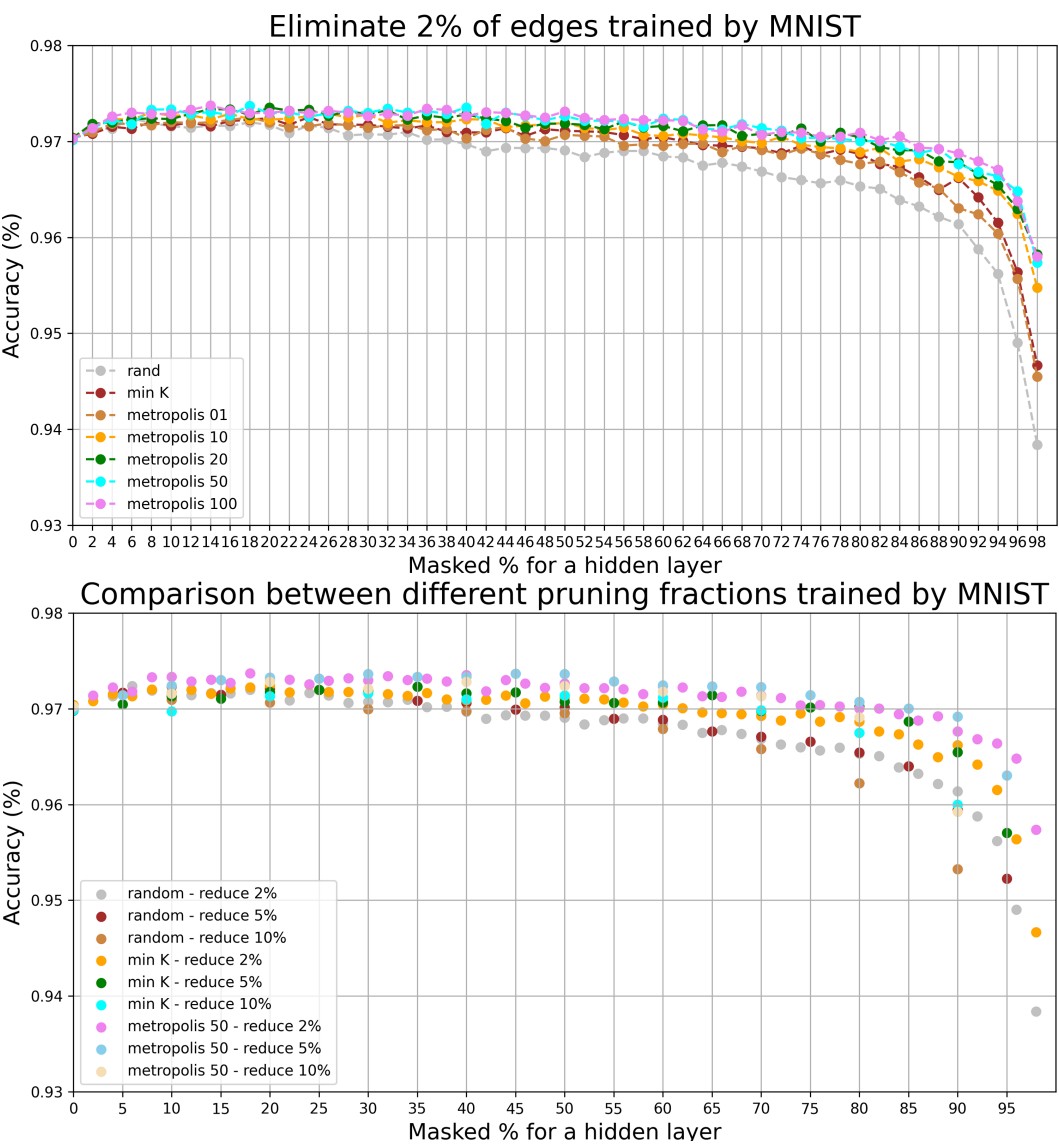

Figure 7: The trends of network performance being pruned at various scales gradually. The network is well trained and fine-tuned by MNIST dataset.

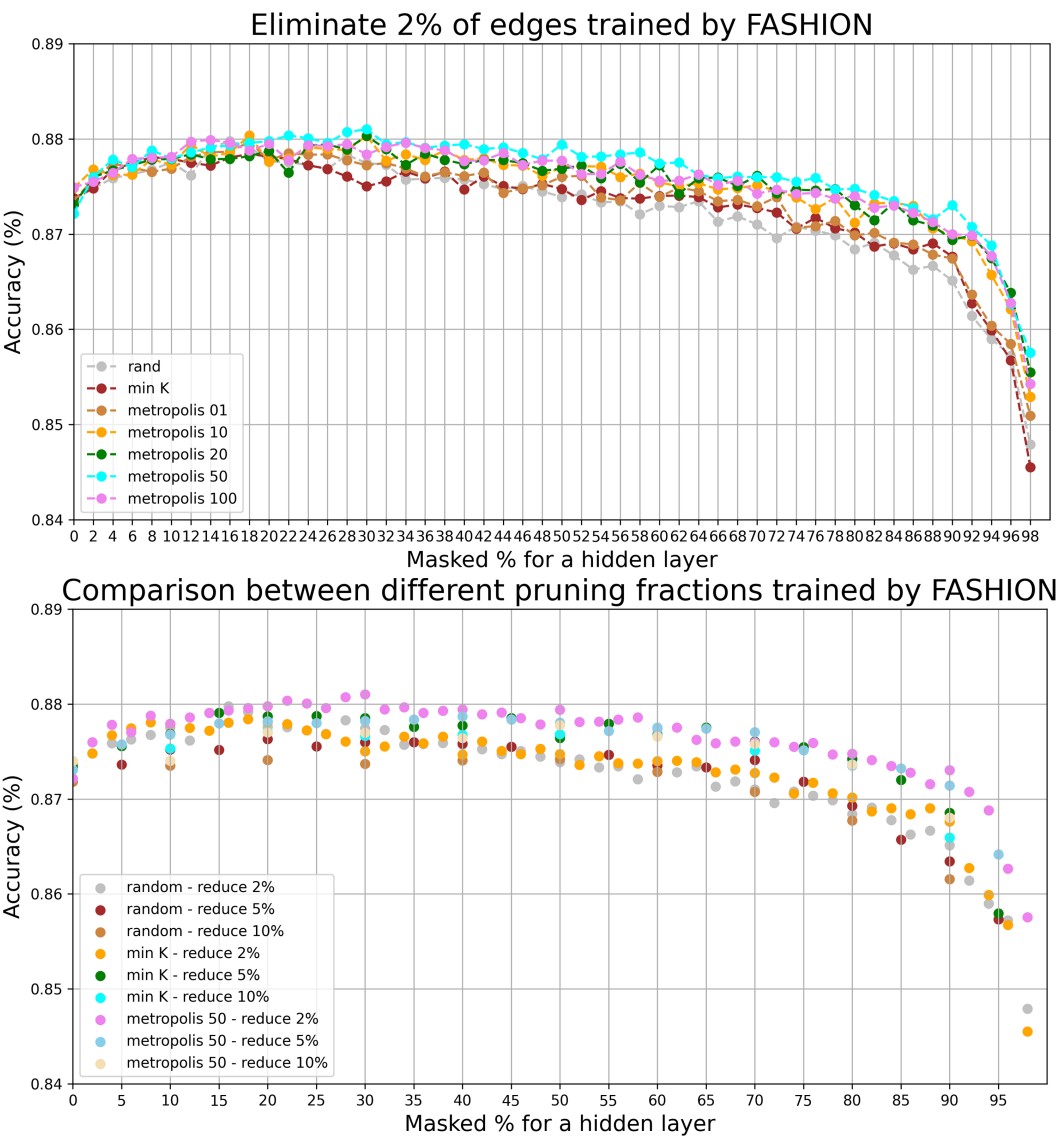

Figure 8: The trends of network performance being pruned at various scales gradually. The network is well trained and fine-tuned by FASHION dataset.

