# OpenReview forum: "Network Pruning Optimization by Simulated Annealing Algorithm"
_ICLR.cc/2022/Conference — ICLR 2022 Submitted_

### Official Review · Reviewer_EBdG · 2021-10-28

**Correctness:** 3
**Technical Novelty And Significance:** 2
**Empirical Novelty And Significance:** 2
**Recommendation:** 3
**Confidence:** 4

**Main Review:**

The idea of using simulated annealing to pruning neural networks seems interesting, but the paper suffers from the issues of 1) insufficient presentation, and 2) limited experiments. My concerns include:
1.	The related work is very sparse. There are many recently published works considering neural architecture search (NAS) and network pruning (NP). But only a few of them is introduced in this study.
2.	The main weakness of this study is the experiment design. The proposed method is only tested on a very simple neural network. Thus, it is difficult to judge whether this method is effective for widely-used convolution neural networks and others.
3.	The performance comparison in this study is also very weak. There are many heuristic algorithms, e.g., evolutionary algorithms, that have been applied on NAS and NP. The proposed method should be compared with those SOTA ones.

**Summary Of The Paper:**

  This paper presents a novel simulated annealing method for pruning neural networks. Both one-shot pruning and gradual pruning strategies are discussed in this study. Experiments are conducted based on a simple neural network structure with only two hidden layers. The results show the effectiveness of the proposed method.

**Summary Of The Review:**

 Although the idea seems interesting, this paper is not technically sound. Experimental results cannot fully support the main conclusion of this paper. Without extensive performance comparison, it is also hard to judge whether this new method can make a real contribution to the related community.

---

> ### Author Response · Authors · 2021-11-17
> **Differences between NAS (macro scale optimization) and our methods (micro scale optimization).**
>
> 1.	The aim of research is to show that pruning does not necessarily lead to an optimal network structure. In that sense, we differ from most NAS approaches which model the macro network choice problem and NP which aim at effective pruning without optimality consideration of the resulting structure. Hence, we only mentioned those highly related and classic methods in our papers to explain the differences of our proposed method, which is to not only prune the network, but also optimize the architecture.
> 2.	We would like to evaluate the effectiveness of our method to a single layer so that factors other than structure optimization such as the relationship between different layers do not obscure the actual observation we want to make on the structure search. Once clarified the dynamics and potentials of our method extending our work to deeper networks will be the next focus of research.
> 3.	The existing heuristic algorithms were applied to only prune a network. However, our annealing algorithm is not only pruning a network, it also optimizes the architecture in order to get better performance. There are two baselines included in this paper, which are random and minimum weight strategies.

---

> > ### Comment · Reviewer_EBdG · 2021-11-26
> > **Comments Added After Reading Author Response**
> >
> > Thanks for answering some of my questions, I appreciate it. But it seems that no substantial improvement has been made in the paper, so I keep my score for this paper.

---

### Official Review · Reviewer_1vEQ · 2021-11-01

**Correctness:** 2
**Technical Novelty And Significance:** 2
**Empirical Novelty And Significance:** 2
**Recommendation:** 1
**Confidence:** 4

**Main Review:**

Strengths
* Interesting application of simulated annealing to solve the network compression problem.
* Detailed overview and motivation for pruning algorithms and simulated annealing.

Weaknesses
* The limited scale and scope of the experiments puts the significance of the results into question. Experiments are conducted on MNIST and FashionMNIST and appear to be single seed, with no clear comparisons to baselines. This is a much smaller scale than similar research published e.g. lottery ticket hypothesis papers. Conducting more detailed comparison to baselines would greatly strengthen the paper.
* The organization of the paper could be improved. There is too much space spent on discussing related work and details of the simulated annealing algorithm, which can be deferred to the appendix. This leaves less room for detailing the actual method and experiments, which come off less clearly as a result. It took me some time to understand that one-shot and gradual pruning
* There are minor grammatical papers throughout the paper.

**Summary Of The Paper:**

This paper explores the application of simulated annealing to network pruning. In the proposed algorithm, a fully trained network has some percentage of weights erased and its connections are perturbed to new configurations, with acceptance happening according to a Boltzmann distribution. Two variations of the algorithm are proposed, one where the target percentage of connections is removed in one-shot, the other where the process is gradual.

**Summary Of The Review:**

I recommend rejection of the paper. Simulated annealing is not a new technique, and the limited scale of the experiments and lack of comparisons to baselines limit the utility of this work in its current form.

---

> ### Author Response · Authors · 2021-11-17
> **Paragraph adjustment**
>
> 1.	Getting the results given from more baselines is our next step. The advantage of starting an experiment from a small network is that we can better observe the effectiveness of our method to one single layer in a network.
> 2.	Thanks for your notice. We will refine the main body of this paper to be more compact.
> 3.	We will do a careful check. Since one of the authors is an Editor in Chief, this problem will be rectified readily.

---

> > ### Comment · Reviewer_1vEQ · 2021-11-29
> > **Maintaining score**
> >
> > Thank you for the response. I agree that observing the effectiveness in a smaller setting is a prerequisite to larger scale experiments. Since these changes in experimental design are not incorporated (and nor should we should expect them to be in the revision period), I will maintain my score.

---

### Official Review · Reviewer_t58P · 2021-11-01

**Correctness:** 3
**Technical Novelty And Significance:** 2
**Empirical Novelty And Significance:** 2
**Recommendation:** 3
**Confidence:** 4

**Main Review:**

The SA presented in the paper is easy to understand and well studied in traditional optimization community. It has proven convergence to optimality. Applying it for network pruning is certainly an interesting idea. The experiment results demonstrate the algorithm can prune up to 80% of the links without too much accuracy deterioration. The algorithm should be not difficult to implement in practice. I believe it helps small neural networks.

The paper has following weaknesses:
1. The algorithm drops or adds a link a time, it can take very long to prune a large network. Why not drop/add a set of links so that the state neighborhood can be larger for large networks?
2. Some content in Appendix should be included in main paper, such as Algorithm 1. Appendix is not guaranteed to be reviewed.
3. Need more baseline comparison. How good is it when comparing with the deep compression (Han et a, 2016, Chen et al, 2020)?
4. There are other network prune studies, for instance ASAP: Architecture Search, Anneal and Prune, by Asaf Noy, etc. What is the difference with them?







**Summary Of The Paper:**

The paper proposes a simulation annealing (SA) algorithm to prune neural network by randomly dropping and adding links according to SA's acceptance-rejection criteria. Experiment results shows the algorithm is useful in reducing number of weights while maintain model accuracy.

**Summary Of The Review:**

The proposed SA method is interesting, however, I am not convinced it can scale well, and the experiments are using small networks. Need more baseline comparison to show its advantages.

---

> ### Author Response · Authors · 2021-11-17
> **The reason why we started from a small network**
>
> 1.	We have actually run simulations also for bigger changes in every step and we can cinclude them in the final paper. The observation is that the performance starts dropping for 5 and more branches at a time since the neighbor structure of the random walk gets far more diverse with moves possibly missing minima. It also affects the stationarity of the Markov chain.
> 2.	We will try to compress the main body so that some contents in appendix can be included.
> 3.	Instead of simply pruning and compressing the network in the sense of data compression/source coding, we tried to focus on architecture optimization. Therefore, the results given from deep compression is not totally comparable with ours. There are 2 baselines we proposed in the paper, which are random pruning and minimum weight pruning methods.
> 4.	As we mentioned in the related work section, NAS tries to determine the best combination of operation blocks that leads to highest performance without overfitting at macro scale, while our method tries to find  the best architecture which is slimmer and has a better performance at micro scale. Our approach can be classified under ASAP methods, not NAS methods.

---

### Official Review · Reviewer_inN3 · 2021-11-03

**Correctness:** 2
**Technical Novelty And Significance:** 2
**Empirical Novelty And Significance:** 1
**Recommendation:** 3
**Confidence:** 4

**Main Review:**

The topic treated in this paper is interesting and important. However, the weaknesses of this paper and the reviewer's concerns are as follows:

- Many pruning algorithms have been developed so far, including pruning methods based on black-box optimization like simulated annealing. The novelty of this paper is not apparent compared to the existing pruning methods.

- The effectiveness of simulated annealing for network pruning is not apparent. For example, a comparison to other black-box optimization methods, such as genetic algorithms, should be conducted to claim the effectiveness of the simulated annealing. Also, the state-of-the-art pruning methods should be included in the baseline.

- Random pruning should be included as the simpler baseline.

- In the experiment, the size of the neural network is too small compared to the current trend. The convolutional neural networks, such as VGG and ResNet, should be considered as the baseline networks to be pruned. The following paper is useful for experimental design.

Davis Blalock, Jose Javier Gonzalez Ortiz, Jonathan Frankle, John Guttag, "What is the state of neural network pruning?," arXiv:2003.03033 (2020).

**Summary Of The Paper:**

This paper develops a method for neural network pruning using simulated annealing. In the proposed method, the mask matrix indicating the pruning location is optimized by simulated annealing. The proposed method is evaluated on pruning tasks of fully connected neural networks in image classification.

**Summary Of The Review:**

This paper proposes the network pruning method using simulated annealing. The basic framework of the proposed method is the same as the usual pruning methods. The experimental comparison with other pruning methods is not performed. Therefore, the reviewer can recognize neither the novelty and effectiveness of the proposed method.

---

> ### Author Response · Authors · 2021-11-17
> **Replies of methods comparison and baseline**
>
> •	 The point of research is to show that pruning does not necessarily lead to an optimal network structure. Vast majority of existing pruning approaches do not consider any optimality criterion, while our approach aims at finding the optimal structure for a partially connected network.
> •	Because simulated annealing is used to optimize the architecture, it is not comparable with the baseline of the state-of-the-art pruning methods that were purely applied to reduce the size of a network.
> •	Random pruning has already been included in our experimental session. The data has been visualized in appendix figure 5~8.
> •	We chose a small network on purpose to observe the dynamics of the algorithm clearly. In a deeper network, there will be various other issues to analyse such as the intertalk between different layers which will be the next thing to study in our research. You can appreciate that discussing a new method on a deep network of 18 layers would not give the same clarity to our discussion.

---

> > ### Comment · Reviewer_inN3 · 2021-11-27
> > **Comments after author replies**
> >
> > Thank you for your responses. I have checked the result of the comparison with random pruning. After re-training, the advantage of the proposed method against random pruning is not apparent. Additionally, I am still concerned that comparison to other black-box optimizers is needed to show the effectiveness of simulated annealing. Therefore, I keep my original score.

---

### Decision · Program_Chairs · 2022-01-20

**Decision:**

Reject

**Comment:**

This paper presents the use of Simulated Annealing (SA) for pruning and optimizing the architecture of a neural network. After reviewing the paper and taking into consideration of the reviewing process, here are my comments:
- The contribution of the paper and the novelty is limited and not well presented
- The related work is very sparse. It requires a major improvement.
- The main concern is about the simplistic experiments and the lack of comparison between the results of the proposal and the SOTA methods.
- Conclusions are not well supported by the results.
From the above, the paper does not fulfill the standards of the ICLR.